# Modeling the Impact of Vaccination on COVID-19 and Its Delta and Omicron Variants

**DOI:** 10.3390/v14071482

**Published:** 2022-07-06

**Authors:** Jianbo Wang, Yin-Chi Chan, Ruiwu Niu, Eric W. M. Wong, Michaël Antonie van Wyk

**Affiliations:** 1Department of Electrical Engineering, City University of Hong Kong, Hong Kong, China; jianbowang11@fudan.edu.cn (J.W.); ycc39@cam.ac.uk (Y.-C.C.); mavanwyk@gmail.com (M.A.v.W.); 2School of Computer Science, Southwest Petroleum University, Chengdu 610500, China; 3Institute for Manufacturing, University of Cambridge, Cambridge CB3 0FS, UK; 4College of Mathematics and Statistics, Shenzhen University, Shenzhen 518060, China; nrw@szu.edu.cn; 5School of Electrical and Information Engineering, University of the Witwatersrand, Johannesburg 2000, South Africa

**Keywords:** COVID-19, SVEIHR model, compartmental model, vaccination, Delta variant, Omicron variant

## Abstract

Vaccination is an important means to fight against the spread of the SARS-CoV-2 virus and its variants. In this work, we propose a general susceptible-vaccinated-exposed-infected-hospitalized-removed (SVEIHR) model and derive its basic and effective reproduction numbers. We set Hong Kong as an example and calculate conditions of herd immunity for multiple vaccines and disease variants. The model shows how the number of confirmed COVID-19 cases in Hong Kong during the second and third waves of the COVID-19 pandemic would have been reduced if vaccination were available then. We then investigate the relationships between various model parameters and the cumulative number of hospitalized COVID-19 cases in Hong Kong for the ancestral, Delta, and Omicron strains. Numerical results demonstrate that the static herd immunity threshold corresponds to one percent of the population requiring hospitalization or isolation at some point in time. We also demonstrate that when the vaccination rate is high, the initial proportion of vaccinated individuals can be lowered while still maintaining the same proportion of cumulative hospitalized/isolated individuals.

## 1. Introduction

The COVID-19 pandemic has persisted for nearly three years since the end of 2019, resulting in huge human and socioeconomic costs. Since the discovery of the original SARS-CoV-2 virus in late 2019, many measures have been taken in an attempt to regain control over the pandemic, including vaccine development and deployment, changes to healthcare systems operations, and non-pharmaceutical interventions (NPIs) such as mask-wearing, social distancing, and, in extreme situations, lockdowns. However, in the meantime, the original virus has mutated to become even more infectious. In particular, the Delta variant became dominant in most of the world soon after its emergence, only to be replaced by the even more transmissable and virulent Omicron variant.

Mathematical modeling [1,2,3] is therefore required to capture the dynamics of the COVID-19 pandemic in the presence of vaccinations, NPIs, and new disease variants. Important and urgent questions that epidemiological modeling can help address include:whether herd immunity can be achieved and the pandemic eventually eradicated;the risk of COVID-19 epidemic resurgence [4];what are the driving socio-demographic and health factors behind the epidemic progression;how to optimize vaccination strategy [5], or whether long-term co-existence with the virus should be pursued instead;whether and when to reopen international or national (e.g., state or provincial) borders;whether additional surge capacity is required at hospitals; andwho to prioritize during vaccination (e.g., providing second doses or booster shots versus increasing efforts to give more people their first dose).

In the following subsection, we summarize the existing literature on epidemiological modeling of COVID-19 in the presence of vaccination.

### 1.1. Background and Related Work

A common approach for epidemiological modeling is the use of compartmental models, where each compartment represents a possible state of individuals in the population. The dynamics of the number of individuals in each compartment can then be described as a system of differential equations. A key concept in epidemiological modeling is the reproduction number, defined as the average number of individuals an infected individual will infect. In particular, the *basic reproduction number* $R_{0}$ assumes a fully susceptible population, whereas the *effective reproduction number* changes over time in accordance with the disease dynamics. In general, a reproduction number less than 1 is associated with herd immunity, in which the disease will eventually die out.

If the basic reproduction number of a disease exceeds one, vaccination can be used to reduce the effective reproduction number. The *vaccination threshold*, defined as the minimum proportion of individuals to be vaccinated in order to reach herd immunity, is generally defined as
(1)$$\rho_{\min}=\frac{1-1/R_{0}}{v_{e}}$$
where $v_{e}$ is the mean vaccine efficacy. However, a problem with this classical formula is that it fails to describe the dynamic interactions *during* an epidemic between disease transmission and ongoing vaccination. This is important for novel diseases, such as COVID-19, for which vaccines are initially not available.

Matrajt et al. [6] proposed a multi-dose vaccination model for COVID-19 and found that if the single-dose efficacy of a vaccine against COVID-19 is high, then prioritizing the first dose, even at the possible expense of delaying the second dose of two-dose vaccines, may help to contain the pandemic more quickly. Meanwhile, [7,8,9,10] consider optimal prioritization of vaccination groups (based primarily on age). However, [6,9] both consider relatively low basic reproduction numbers ($R_{0}$) for COVID-19, even when all NPIs are relaxed, compared to current estimates of $R_{0}\approx 5$ for the Delta variant [11] and $R_{0}\approx 9.5$ for the Omicron variant [12]. Furthermore, [7,8] do not provide estimates of herd immunity thresholds for COVID-19.

Giordano et al. [13] show that NPIs have a higher effect than vaccination alone and advocate for the need to keep NPIs in place during the first phase of the vaccination campaign. However, the authors do not consider the relaxation of NPIs later on during the pandemic, when the vaccination ratio is high. We shall consider this latter scenario in this paper. In contrast, [14] considers the optimization problem of minimizing the total impact of social distancing measures over time given certain healthcare capacity constraints. Therefore, the gradual relaxation of NPIs occurs naturally as vaccination ratios increase.

Finally, a shared feature of [7,8,9] is that they all employ complex compartmental models, with stratification based on age and possibly other factors such as vaccination status. In this paper, we instead use a much simpler model, without the need for many stratified compartments, allowing for easier sensitivity analysis and interpretation of the results. Although many conclusions and recommendations (such as the prioritization of certain groups for vaccination) can only be obtained using stratified models, we demonstrate that some important insights can still be obtained using our simpler model.

### 1.2. Contributions of This Paper

The main contributions of this paper are as follows:We propose a mathematical model to characterize the epidemiological process of COVID-19 with vaccination and derive formulas for its vaccine reproduction number (similar to $R_{0}$ but assuming a fully vaccinated population) and effective reproduction number. We show that, based on the current ratio of CoronaVac (Sinovac) and Comirnaty (Pfizer–BioNTech) vaccinations in Hong Kong, herd immunity could not have been obtained via vaccination alone. Furthermore, no ratio of CoronaVac versus Comirnaty vaccinations can achieve herd immunity against Omicron via vaccination alone.Using our new model, we observe the impact of vaccination in terms of various parameters (including the initial vaccination coverage at the start of a new outbreak, the rate of new vaccinations, the average vaccine efficacy, and NPI intensity) on the cumulative number of hospitalized cases, for both the original and Delta strains of the SARS-CoV-2 virus, as well as the Omicron variant.We compute, for different vaccination rates and vaccine efficacies, the minimum initial vaccination coverage required such that the cumulative number of hospitalized/isolated cases is less than a given percentage of the population. Furthermore, we demonstrate how a high vaccination rate decreases the required initial vaccination coverage.We find that the traditional formula (Equation 1) for herd immunity, assuming a static vaccination coverage (i.e., an initial vaccinated group with no additional vaccinations), corresponds to a cumulative hospitalization ratio of about one percent.We show how regions can achieve herd immunity at a lower vaccination coverage ratio than that predicted by (Equation 1), but at the cost of additional infections (*natural immunity* versus *induced immunity*).

Based on prevailing Hong Kong government policy during the Delta and previous waves, we assume that all infected cases are sent to hospital or isolation in an ancillary facility, even when infeasible due to lack of actual healthcare capacity (e.g., during the Omicron wave).

Note that our model does not differentiate between hospitalization or isolation in a non-hospital setting, as both outcomes prevent further transmission to the general population. Although we take the population of Hong Kong as an example in this paper, our model is general and thus applicable to any approximately homogeneous region. A previous version of our proposed model, without vaccination, has been applied to several different regions and found to be quite accurate at modeling the evolution of the COVID-19 pandemic. In this regard, it is expected that the model used in this paper can be also applied to other cities and countries for capturing the dynamics of vaccination.

In summary, rather than focusing on the conditions required to achieve herd immunity, we focus instead on the cumulative number of infections throughout the course of an epidemic. One justification for this is that the definition of herd immunity does not discriminate between vaccine-induced immunity and natural immunity from disease recovery. Although it is theoretically possible to achieve herd immunity via natural immunity alone, due to the severity of some COVID-19 cases and the limitations on healthcare capacity, this is not a desirable approach.

## 2. Methods

### 2.1. Compartmental Modeling for COVID-19

In [15,16], an SEIHR model is proposed for the modeling of COVID-19 in Hong Kong and other regions in the absence of vaccination. The model was found to accurately describe the number of confirmed cases in each region over time. Compared to the classical SEIR model [2]—which contains a susceptible (S), exposed (E), infected (I), and removed (R) compartment—the SEIHR model adds a “hospitalized” (H) compartment for confirmed cases in hospital (*or non-clinical isolation*) and makes the E compartment contagious to model the effect of asymptomatic and pre-symptomatic infections. By modeling both unconfirmed cases (using the E compartment) and the isolation of confirmed cases (using the H compartment), the SEIHR model achieves better results than the classical SIR and SEIR models when fitting actual COVID-19 infection data.

In this paper, we modify the previous SEIHR model by adding a *“vaccinated” (V)* compartment. We call this new model the **SEIVHR** model. For simplicity, we assume a single-dose vaccine where vaccination provides immediate protection, with an efficacy of $v_{e}$. We also assume that the rate of vaccinations at time *t* is proportional to S(t), the number of unvaccinated susceptible individuals at time *t*. The full list of model parameters is given in Table 1, along with some additional notation. The system of differential equations describing the SVEIHR model can therefore be written as follows:(2)$$\begin{array}{cc} X\left(t\right) & ={\left[\begin{array}{cccccc} S(t) & V(t) & E(t) & I(t) & H(t) & R(t) \end{array}\right]}^{T}\\ \frac{d}{dt}X\left(t\right) & =\overset{G}{\overbrace{\left[\begin{array}{ccccc} -k-\eta (t) & 0 & 0 & 0 & 0\\ k & -\eta_{V}\left(t\right) & 0 & 0 & 0\\ \eta (t) & \eta_{V}\left(t\right) & -\beta -\delta & 0 & 0\\ 0 & 0 & \beta & -\gamma -\delta & 0\\ 0 & 0 & 0 & \gamma & -\delta \\ 0 & 0 & \delta & \delta & \delta \end{array}\right]}}\left[\begin{array}{c} S(t)\\ V(t)\\ E(t)\\ I(t)\\ H(t) \end{array}\right], \end{array}$$
where
$$\begin{array}{cc} \eta (t) & =\varepsilon \frac{E(t)}{N}+\alpha \frac{I(t)}{N}\\ \eta_{V}\left(t\right) & =\varepsilon_{V}\frac{E(t)}{N}+\alpha_{V}\frac{I(t)}{N} \end{array}$$
where Y(t) denotes the number of individuals in compartment Y at time *t* (for Y in $\left\{S,V,E,I,H,R\right\}$), and α, ε, γ, and δ are all positive.

**Remark** **1.**
*The non-diagonal elements of G correspond to the transition rates shown in Figure 1, whereas the diagonal elements are defined such that the column sums of G are all zero.*


**Remark** **2.**
*The transmission rates η(t) and $\eta_{V}\left(t\right)$ are the weighted averages of the individual transmission rates for the population, where the individual transmission rate of individuals outside of the E and I compartments is zero. These individual transmission rates depend on whether the target individual is vaccinated or not.*


**Definition** **1.**
*The vaccine efficacy $v_{e}$ is the reduction in transmission rate for vaccinated targets, such that $\varepsilon_{V}=\varepsilon \left(1-v_{e}\right)$ and $\alpha_{V}=\alpha \left(1-v_{e}\right)$. In other words, $v_{e}=0$ denotes no difference between the S and V compartments, whereas $v_{e}=1$ denotes that *
*
**no**
*
* individuals in compartment V will ever catch the disease.*


### 2.2. Reproduction Numbers

Recall that for the original SEIHR model proposed in [16],
(3)$$R_{0}=\frac{\varepsilon }{\beta +\delta }+\frac{\alpha \beta }{\left(\gamma +\delta \right)\left(\beta +\delta \right)}$$
and herd immunity can be reached if $1-1/R_{0}$ of the population achieves immunity, either via vaccination or disease recovery. We define the following numbers:The *basic reproduction number* $R_{0}$, referred to above, is the average number of individuals an infectious individual will infect in a wholly susceptible population, without vaccination.The *vaccine reproduction number* $R_{V}$ is the average number of individuals an infectious individual will infect in a wholly vaccinated population.The *effective reproduction number* $R_{e}\left(t\right)$ is the average number of individuals an infectious individual will infect based on the current situation at time *t*.

$R_{V}$ can be obtained from (Equation 3) by simply replacing the transmission rate parameters ε and α by their counterparts for vaccinated individuals:(4)$$R_{V}=\frac{\varepsilon_{V}}{\beta +\delta }+\frac{\alpha_{V}\beta }{\left(\gamma +\delta \right)\left(\beta +\delta \right)}.$$

By extension, the effective reproduction number can be derived as a linear combination of $R_{0}$ and $R_{V}$:(5)$$R_{e}\left(t\right)=\frac{S(t)}{N}R_{0}+\frac{V(t)}{N}R_{V}.$$

**Remark** **3.**
*The effective reproduction number $R_{e}\left(t\right)$ is a weighted average of the basic and vaccine reproduction numbers, where the weights are the proportion of unvaccinated (S) and vaccinated (V) susceptible individuals in the population.*


If NPIs are introduced with a control intensity of *c*, 0≤c≤1, then the transmission rates α, ε, $\alpha_{V}$, and $\varepsilon_{V}$ and the reproduction numbers $R_{V}$ and $R_{e}\left(t\right)$ are all scaled by a factor of 1−c and written as:$$\begin{array}{cc} R_{V,c} & =\left(1-c\right)R_{V}\\ R_{e,c}\left(t\right) & =\left(1-c\right)R_{e}\left(t\right). \end{array}$$

### 2.3. Herd Immunity and Multiple Vaccines

For regions where multiple vaccines have been administered, we can use the average efficacy across all vaccines, weighted by the proportion of vaccinated individuals taking each vaccine. If the efficacy and uptake ratio of each vaccine *i* are denoted as $v_{e,i}$ and $r_{i}$, respectively, then the average efficacy is
(6)$$v_{e}=\sum_{i}v_{e,i}r_{i}.$$

In Hong Kong, two vaccines have been administered, namely Comirnaty (Pfizer–BioNTech) and CoronaVac (Sinovac). As of 10 April 2022, 3,438,310 people in Hong Kong have completed two doses of Comirnaty, and 2,508,108 people have completed two does of CoronaVac [17], giving $r_{1}=0.6$ and $r_{2}=0.4$. Based on efficacy studies for Delta [18,19,20], we obtain $v_{e,1}=0.88$ and $v_{e,2}=0.59$. From these values, we obtain an average vaccine efficacy of $v_{e}=0.758$.

Given an estimated $R_{0}$ of 5 for Delta [11], we obtain a vaccination threshold of
ρmin=1−1/R0ve   R0=5,ve=0.758=1.055>1,
which implies that herd immunity via vaccination alone was impossible in Hong Kong given the current mix of vaccines. However, a greater proportion of Comirnaty vaccinations could have achieved herd immunity via vaccination alone. The minimum value of $r_{1}$ for this scenario, assuming a fully vaccinated population, can be obtained using (Equation 1) and (Equation 6):(7)$$\begin{array}{ccc} \frac{1-1/R_{0}}{v_{e,1}r_{1}^{\min}+v_{e,i}\left(1-r_{1}^{\min}\right)} & =1 & \left(v_{e,1}=0.88,\;v_{e,2}=0.59,\;R_{0}=5\right)\\ r_{1}^{\min} & =0.7241. \end{array}$$

For Omicron, estimates for the basic reproduction number for Omicron range from 7.25 to 11.88 [12]. In this paper, we shall assume an $R_{0}$ value of 9.5 for exemplary purposes. In other words, even with a fully vaccinated population, the required minimum vaccine efficacy would be $1-1/R_{0}=0.894$. In contrast, efficacy studies [21] have given values of $v_{e,1}=0.655$, considerably less than the required threshold, whereas for CoronaVac, Yale researchers found effectively *no* neutralizing antibodies in volunteers against the Omicron variants even after two doses administered [22]. Therefore, herd immunity via vaccination alone is impossible against Omicron.

More generally, we can state the following:If $1-1/R_{0}\leq \min\left(v_{e,1},v_{e,2}\right)$: either vaccine can achieve herd immunity via vaccination alone.If $1-1/R_{0}>\max\left(v_{e,1},v_{e,2}\right)$: neither vaccine can achieve herd immunity via vaccination alone.Otherwise: whether herd immunity can be achieved via vaccination alone depends on the mix of vaccines administered.

Where herd immunity is impossible via vaccination alone, it can instead be achieved via a mix of vaccination and natural immunity (i.e., via the recovery of infected individuals).

### 2.4. Asymptotic Behavior

Let $R\left(\infty \right)=\lim_{t\to \infty }R\left(t\right)$ denote the limiting number of removed (recovered or deceased) individuals in system (Equation 2), and let $H_{c}\left(\infty \right)$ denote the cumulative number of hospitalized/isolated individuals as t→∞. We now explore the relationship between R(∞) and $H_{c}\left(\infty \right)$. First, we note that there are three paths from compartment E to compartment R, namely E→R, E→I→R, and E→I→H→R. Let the proportion of exposed individuals that follow each path be $p_{1}$, $p_{2}$, and $p_{3}$, respectively.

Since all paths from E lead to R, and all paths to R include E, the cumulative number of exposed individuals, $E_{c}\left(\infty \right)$, is equal to R(∞). Furthermore, since compartment H exists on the E→I→H→R path only, we obtain $H_{c}\left(\infty \right)=p_{3}R\left(\infty \right)$. Finally, from Figure 1, we can see that
(8)$$p_{3}=\left(\frac{\gamma }{\gamma +\delta }\right)\left(\frac{\beta }{\beta +\delta }\right).$$

**Remark** **4.**
*The value $p_{3}$ in (Equation 8) denotes the proportion of exposed individuals who are hospitalized or isolated for the disease. In other words, $1-p_{3}$ of cases recover on their own without being detected.*


## 3. Results and Discussions

In this section, we present simulation results for our SVEIHR model under various scenarios. For the sake of example, we set N= 7,394,700 [23], the approximate population of Hong Kong as of 12 August 2021.

### 3.1. Ancestral Strain

Based on [16], we set (ε,α,β,γ,δ)=(0.48,0.5,0.14,1,0.1) for the ancestral strain of COVID-19, obtaining $R_{0}=2.26$. Figure 2 shows the evolution of each compartment of the SVEIHR model (Equation 2), given initial state X(0)=(N−1,1,0,0,0,0), that is, a single exposed individual at time 0, a vaccination rate of k=0.002, a vaccine efficacy of $v_{e}=0.95$ (approximately that of Comirnaty for the ancestral strain), and no NPIs (c=0). It is shown that the number of hospitalized/isolated individuals mostly dies out within 100 days; however, most of the population will eventually catch the disease in this scenario, as shown by the dark green line. Additionally, a large proportion of cases are undetected, as shown by the large gap between the two green lines. Furthermore, the number of successful vaccinations (vaccinated individuals that do not later become exposed) remains low, at about one million at t=100.

For the ancestral strain, we define individual vaccine efficacies of $v_{e,1}=0.95$ and $v_{e,2}=0.51$ and vaccine uptake ratios of $r_{1}=0.6$ and $r_{2}=0.4$ for the Comirnaty and CoronaVac vaccines, respectively. Applying (Equation 6), we obtain a mean vaccine efficacy of $v_{e}=0.774$. Next, applying (Equation 1), the theoretical vaccination threshold for herd immunity is (1−1/2.26)/0.774=0.7203, implying that herd immunity against the ancestral strain was possible via vaccination alone. However, as we will show in this paper, this would have required a high ratio of vaccination individuals *before* an outbreak occurred, which was impossible as vaccines had yet to be developed when the ancestral strain first propagated.

### 3.2. The Second and Third Waves in Hong Kong

We now explore how the second and third waves in Hong Kong would have evolved differently under vaccination, using parameters for the ancestral strain ($v_{e}=0.774$). The cumulative number of hospitalized/isolated cases over time is shown in Figure 3 and Figure 4 for different initial vaccination coverage ratios $r_{V}\left(0\right)$ and vaccination rates *k*, with initial model states as listed in Table 2.

The results demonstrate that vaccination has a significant impact in reducing the $H_{c}\left(\infty \right)$ for both waves, despite the relatively lower efficacy of CoronaVac, which was chosen by approximately two-fifths of the population. In fact, this holds true even in the case of $r_{V(0)}=0$, denoting zero initial vaccinations, with vaccination commencing only at the start of the disease wave. The effect is especially pronounced when the vaccination rate *k* is large. The reduction in cumulative hospitalizations increases when the vaccination rate and/or initial vaccination coverage is increased.

### 3.3. Sensitivity with Respect to Initial Vaccination Coverage $r_{V}\left(0\right)$
and Vaccination Rate *k*

In the following subsections, we use contour plots to investigate $H_{c}\left(\infty \right)$ with respect to pairs of parameters in the SVEIHR model related to vaccination and NPIs. The other parameters of the model are set to (ε,α,β,γ,δ)=(1.3637,1.3637,0.2273,1,0.1) for Delta, thus obtaining a basic reproduction number of $R_{0}=5$, and (ε,α,β,γ,δ)=(2.5767,2.5767,0.2273,1,0.1) for Omicron, thus obtaining $R_{0}=9.5$. This is consistent with the estimates of $R_{0}=5.08$ and $R_{0}=7.25$ to 11.88 obtained in [11] and [12] for Delta and Omicron, respectively.

Figure 5A shows $H_{c}\left(\infty \right)$ with respect to the initial vaccination coverage $r_{V}\left(0\right)$ and vaccination rate *k* for different control intensities *c* and a vaccine efficacy of $v_{e}=0.88$, as for Comirnaty [18]. Figure 5B shows the same, but with a vaccine efficacy of $v_{e}=0.655$, as for CoronaVac [19]. The results show that relatively low numbers of hospitalizations are possible under Comirnaty-only vaccination, even without NPIs for control, whereas a high control intensity is required for CoronaVac-only vaccination, regardless of the number of vaccinations. This is consistent with (Equation 7), where it is demonstrated that the minimum ratio of Comirnaty vaccinations to achieve herd immunity via vaccination alone is $r_{1}^{\min}=0.724$, based on the static formula for herd immunity.

However, for the Omicron variant, Figure 5C,D show that regardless of vaccine, strong NPI control (c=0.75) is necessary to ensure a low number of hospitalizations or isolations. This is due to both the increased transmission rate of Omicron (higher $R_{0})$ and a lower vaccine efficacy. However, Comirnaty still produces lower hospitalization numbers than CoronaVac.

Figure 5 shows that increasing either the NPI control intensity (*c*) or the vaccination rate (*k*) decreases the minimum initial vaccination coverage $r_{V}\left(0\right)$ required for $H_{c}\left(\infty \right)$ to remain under a certain limit. In other words, governments do not need to wait for the theoretical vaccination threshold $\rho_{\min}=\left(1-1/R_{0}\right)/v_{e}$ to be reached to lift NPIs, as long as the vaccination rate is sufficiently fast. On the other hand, even for the less virulent Delta variant and all-Comirnaty vaccination, a relatively high $r_{V}\left(0\right)$ value of 0.75 is still required when c=0 (no NPI controls) to avoid a large outbreak, as shown by the green transition zone in Figure 5A.

### 3.4. Sensitivity with Respect to Vaccine Efficacy $v_{e}$ and Initial
Vaccination Coverage $r_{V}\left(0\right)$

Figure 6 shows $H_{c}\left(\infty \right)$ with respect to the vaccine efficacy $v_{e}$ and initial vaccination coverage $r_{V}\left(0\right)$ for simulated Delta and Omicron outbreaks. It is shown that a very high vaccine efficacy and initial vaccination coverage is required to control the cumulative number of hospitalizations. Note that for k=0, the boundaries of the green transition arc, denoting a set of tipping points separating low and high levels of hospitalization, are around $r_{V(0)}=0.8$ (for perfect vaccine efficacy) and $v_{e}=0.8$ (for perfect initial vaccination coverage), corresponding to the expected value of $1-1/R_{0}$. On the other hand, increasing the vaccination rate *k* elongates the transition arc in the $r_{V(0)}$ axis but not the $v_{e}$ axis. This is because when $r_{V(0)}=1$, the value of *k* has no effect, as all individuals are already vaccinated at the start of the outbreak.

Finally, compared to Delta, the higher $R_{0}$ value of Omicron results in a much smaller region in which the cumulative number of hospitalizations/isolations can be kept low. This reflects the reality where vaccination alone is not sufficient to reach herd immunity against Omicron, given the current state of the available vaccines. Instead, *natural immunity* caused by infection has played a major role in ending the “fifth” wave wave of COVID-19 in Hong Kong (the first caused by Omicron), whereas no previous wave had caused significant case numbers in Hong Kong by comparison.

### 3.5. Measuring the Effective Reproduction Number over Time

Considering the Omicron example from Section 3.4, we plot the effective reproduction number $R_{e}\left(t\right)$ over time for various vaccine efficacy values $v_{e}$ and initial vaccination coverage ratios $r_{V}\left(0\right)$, with a vaccination rate of k=0.002 and no NPIs (c=0). The results, shown in Figure 7, demonstrate that the lowest $v_{e}$ values actually yield the lowest $R_{e}\left(t\right)$ for sufficiently large *t*, despite yielding the highest $R_{e}\left(t\right)$ at the beginning of the outbreak. This can be explained by noting that a low vaccine efficacy results in a very large number of infections at the beginning of the outbreak, causing greater depletion of the susceptible and vaccinated compartments. In other words, a high transmission rate for susceptible individuals does not automatically lead to a high number of new infections if there are few people left to infect.

### 3.6. Sensitivity with Respect to Vaccine Efficacy $v_{e}$ and NPI Control
Intensity *c*

Figure 8 shows $H_{c}\left(\infty \right)$ with respect to $v_{e}$ and the control intensity *c* of NPIs for different $r_{V(0)}$ and a vaccination rate of k=0.005. It is shown that for an initial vaccine coverage of less than 70 percent ($r_{V(0)}\leq 0.7$), even a perfect vaccine ($v_{e}=1$) cannot control either outbreak (Delta or Omicron) without some NPI control. For reference, [24] estimates that mask wearing alone can achieve a control intensity of c=0.2 (i.e., 80% efficacy) “among compliant subjects”.

For Omicron, the blue region (representing a relatively low number of hospitalized/isolated individuals) is much smaller than for Delta, which again confirms that Omicron is much more infectious than Delta. This explains why more than one million people in Hong Kong were infected by Omicron despite stringent NPI controls and high vaccine availability, whereas the previous non-Omicron waves in Hong Kong caused significantly fewer infections in comparison.

### 3.7. Sensitivity with Respect to Vaccine Efficacy $v_{e}$ and Reproduction
Number $R_{0}$

Estimates of the basic reproduction number for Omicron range from 7.25 to 11.88 [12], which is much higher than even that for Delta. In this subsection, we therefore explore the effect of increasing $R_{0}$ on the cumulative number of hospitalizations/isolations, by increasing the transmission rates α and ε in fixed proportion. Figure 9 shows $H_{c}\left(\infty \right)$ with respect to $v_{e}$ and $R_{0}$ for various values of $r_{V(0)}$, a vaccination rate of k=0.002, and an NPI control intensity of c=0.8. The results demonstrate that the minimum vaccine efficacy required to avoid a large outbreak increases with $R_{0}$, until a certain threshold is reached, upon which the outbreak cannot be controlled via vaccination alone. This threshold increases with $r_{V(0)}$, implying that a more transmissible disease requires a larger initial vaccination coverage ratio or stricter NPIs to prevent a large outbreak.

Figures for c=0.2 and 0.5 are provided in the Appendix A. The results show that increasing the control intensity *c* increases the maximum $R_{0}$ of the disease for which a low number of hospitalizations/isolations can be maintained (for the same vaccination efficacy $v_{e}$).

### 3.8. Dynamic Versus Static Vaccination Thresholds

We consider, for various vaccine efficacies $v_{e}$, the minimum vaccination threshold required such that the cumulative number $H_{c}\left(\infty \right)$ of hospitalizations/isolations is less than some value *h*. Note that this is different from the theoretical herd immunity threshold defined in (Equation 1). For various vaccination rates *k*, we compute the theoretical herd immunity threshold $\rho_{\min}$, as well as two additional values:$r_{V(0)}^{\min}$ denotes the minimum *initial* vaccination coverage for a given vaccination rate *k* such that $H_{c}\left(\infty \right)<h$.$r_{\mathrm{HI}}$ denotes the *dynamic vaccination threshold*, which we define as $r_{\mathrm{HI}}=V\left(t^{*}\right)/N$, where $V\left(0\right)/N=r_{V(0)}^{\min}$ is the initial vaccination coverage and $t^{*}$ denotes the minimum time after which $H_{c}\left(\infty \right)<h$ *even if* all new vaccinations were to cease at time $t^{*}$.

The results are shown in Figure 10 for $h\in \left\{0.01N,0.1N\right\}$, no NPIs (c=0), and various vaccination rates *k*. As in Section 3.7, the parameters for different values of $R_{0}$ are based on scaling the base parameters for the Delta variant such that the $R_{0}$ value is equal to that shown in the figures. Figures for additional values of *h* are available in the Appendix A.

The results show that when *k* is high, $r_{V(0)}^{\min}$ can be significantly less than $r_{\mathrm{HI}}$ and $\rho_{\min}$, meaning that faster vaccination results in a lower initial vaccine coverage ratio required to maintain the size of an outbreak under a given limit. A consequence of this is that governments can consider lifting NPIs even before the theoretical vaccination threshold for herd immunity is reached. Additionally, for h=0.01N, $r_{\mathrm{HI}}$ and $\rho_{\min}$ are approximately equal, whereas for h=0.1N, $r_{\mathrm{HI}}$ can be significantly less than $\rho_{\min}$. This is because when h/N is large, a significant proportion of individuals are allowed become infected and gain *natural immunity* from the disease in that way, reducing the vaccination requirement to reach herd immunity. However, over-reliance on natural immunity may lead to a large number of hospitalizations and a decline in the quality of medical care due to capacity exceedance, increasing the case fatality ratio.

## 4. Conclusions

In this paper, we proposed an SVEIHR model for COVID-19 to capture the dynamic processes of virus transmission and vaccination, and their interactions. Based on the numerical results in this paper, we present the following conclusions:For the ancestral strain of SARS-CoV-2 ($R_{0}=2.26$) and a 6:4 ratio of Comirnaty and CoronaVac vaccination, as currently observed in Hong Kong, the theoretical vaccination threshold for herd immunity is $\rho_{\min}=0.7203$, implying that herd immunity is possible via vaccination alone.For the Delta variant ($R_{0}=5$), herd immunity is possible via vaccination alone using Comirnaty, but not using the current ratio of Comirnaty and CoronaVac in Hong Kong, even if the *entire* population is to be vaccinated—the proportion of Comirnaty vaccinations has to increase.For the Omicron variant ($R_{0}\approx 9.5$), herd immunity is impossible via vaccination alone, regardless of vaccine mix and even if the *entire* population is to be vaccinated. Similar results can be obtained for newer variants such as BA.4 or BA.5 by adjusting the model’s parameters.NPI control measures are required to limit an outbreak until vaccination has reached a sufficiently high level. Moreover, if the basic reproduction number is high and the vaccine efficacy relatively low, then control measures are required to prevent a mass outbreak even when vaccine coverage is high (in some cases, even if the population is fully vaccinated).Faster vaccination results in a lower initial vaccine coverage ratio required to maintain the size of an outbreak under a given limit. Thus governments can consider lifting controls even before the theoretical vaccination threshold for herd immunity is reached.Increasing the number of individuals who are allowed to become infected reduces the requirement on the number of individuals that need to be vaccinated before herd immunity is reached. However, over-reliance on infection-acquired natural immunity may lead to a large number of hospitalizations and a decline in the quality of medical care due to capacity exceedance, increasing the case fatality ratio.

### Limitations of This Work

There are still many aspects of vaccination dynamics that need further investigation. For example, our model assumes that immunity via vaccination is immediate and does not consider the reduced vaccine efficacy between the first and second doses of two-dose vaccines or in the period immediately after taking the second dose. We also do not consider the waning of vaccine protection over time or the introduction of booster shots (i.e., additional doses of vaccine to enhance protection), the formulations of which may also be altered to provide extra protection against new variants. Studies [25] have shown that vaccine antibody levels start to wane around 2–3 months after injection of the Comirnaty or Vaxzervria (Oxford–AstraZeneca) vaccines; however, further study is required to show the relationship between vaccine antibody levels and protection against COVID-19 infection.

Heterogeneities in the population are also ignored, whereas [26] showed that the herd immunity threshold can be lower in a heterogeneous population than a homogeneous one. We also do not model differences in the efficacy of different vaccines, instead using the average efficacy only, nor do we consider scenarios with multiple disease variants in co-existence, mainly due to the current dominance of Omicron in most of the world. Migration between heterogeneous geographical regions, which may have different vaccination rates, vaccine efficacies, and control measures, is also not considered here. Finally, it may be useful to model not only total “hospitalizations” but also the number of patients requiring various levels of medical care, including non-clinical isolation and observation only, basic treatment, mechanical ventilation, and intensive care.

In future work, we will extend our model to include the effects of heterogeneity with respect to age, social activity, and administered vaccines. Other effects for possible consideration include waning vaccine efficacy over time, severe or critical cases as a subset of hospitalized cases, and the co-evolution of multiple disease strains.

## Figures and Tables

**Figure 1 viruses-14-01482-f001:**
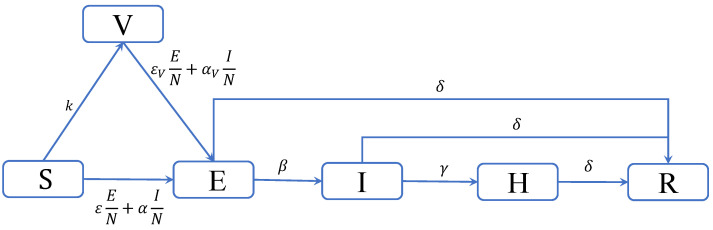
Diagram of the SVEIHR model. Individuals in the model transition between the six states of the model in accordance to the labeled per-individual transition rates between states. The “recovered” (R) state is always a final state, while the “susceptible” (S) and “vaccinated” (V) states become final if there are no longer any “exposed” (E) or “infected” (I) individuals in the system.

**Figure 2 viruses-14-01482-f002:**
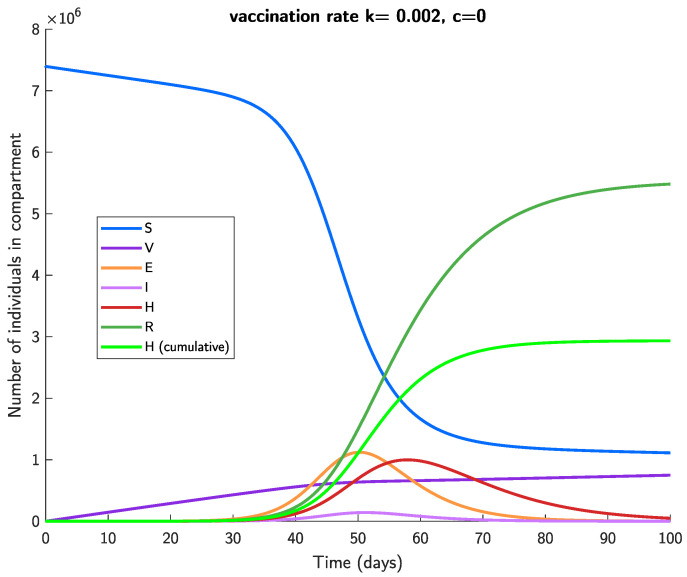
Evolution of the SVEIHR model for the ancestral strain of SARS-CoV-2, with parameters (ε,α,β,γ,δ)=(0.48,0.5,0.14,1,0.1), vaccination rate k=0.002, vaccine efficacy $v_{e}=0.95$, and initial values X(0)=(N−1,1,0,0,0,0). The first six lines in the legend depict the number of individuals in each compartment over time, while the final (bright green) line depicts the cumulative number of hospitalized or isolated individuals.

**Figure 3 viruses-14-01482-f003:**
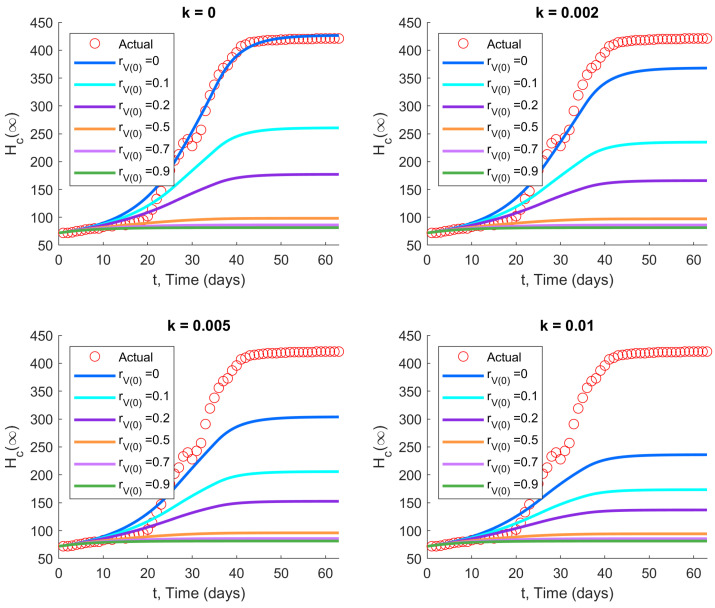
Impact of vaccination with two vaccine types on the second wave of COVID-19 in Hong Kong: (ε,α,β,γ,δ)=(0.48,0.5,0.14,1,0.1) and $v_{e}=0.774$, with initial values as in Table 2. Increasing the initial vaccination coverage ($r_{V(0)}$) and/or the subsequent vaccination rate (*k*) significantly reduces the cumulative number of hospitalizations and isolations.

**Figure 4 viruses-14-01482-f004:**
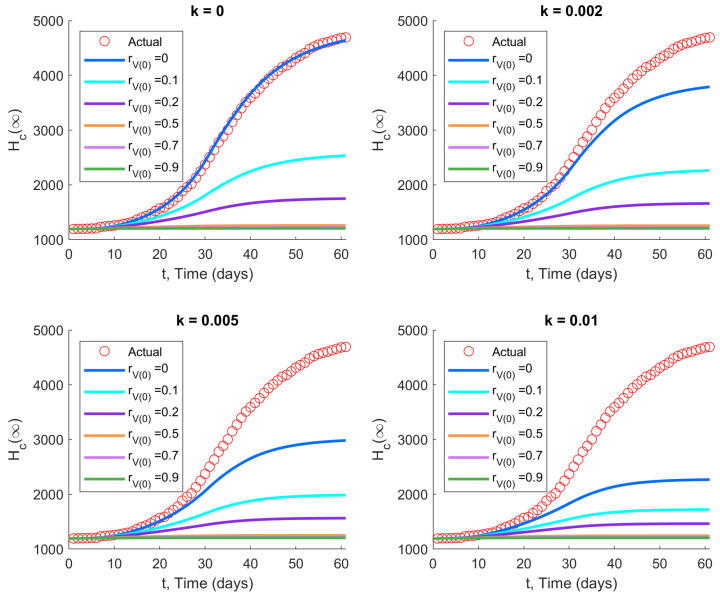
Impact of vaccination with two vaccine types on the third wave of COVID-19 in Hong Kong: (ε,α,β,γ,δ)=(0.48,0.5,0.14,1,0.1) and $v_{e}=0.774$, with initial values as in Table 2. Increasing the initial vaccination coverage ($r_{V(0)}$) and/or the subsequent vaccination rate (*k*) significantly reduces the cumulative number of hospitalizations and isolations.

**Figure 5 viruses-14-01482-f005:**
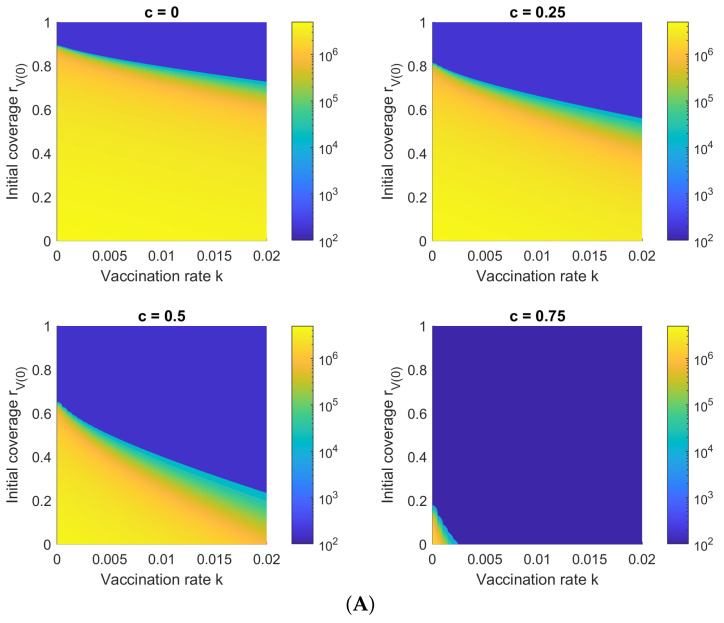
Cumulative hospitalizations/isolations $H_{c}\left(\infty \right)$ with respect to initial vaccination coverage $r_{V}\left(0\right)$ and vaccination rate *k* for (**A**) a simulated Delta outbreak with Comirnaty vaccinations, $v_{e}=0.88$; (**B**) a simulated Delta outbreak with CoronaVac vaccinations, $v_{e}=0.59$; (**C**) a simulated Omicron outbreak with Comirnaty vaccinations, $v_{e}=0.655$; and (**D**) a simulated Omicron outbreak with CoronaVac vaccinations, $v_{e}=0.51$. The results show that while Comirnaty can control a Delta outbreak without NPIs (if vaccination is high), only stringent NPI controls are able to limit the spread of Omicron, regardless of vaccination status.

**Figure 6 viruses-14-01482-f006:**
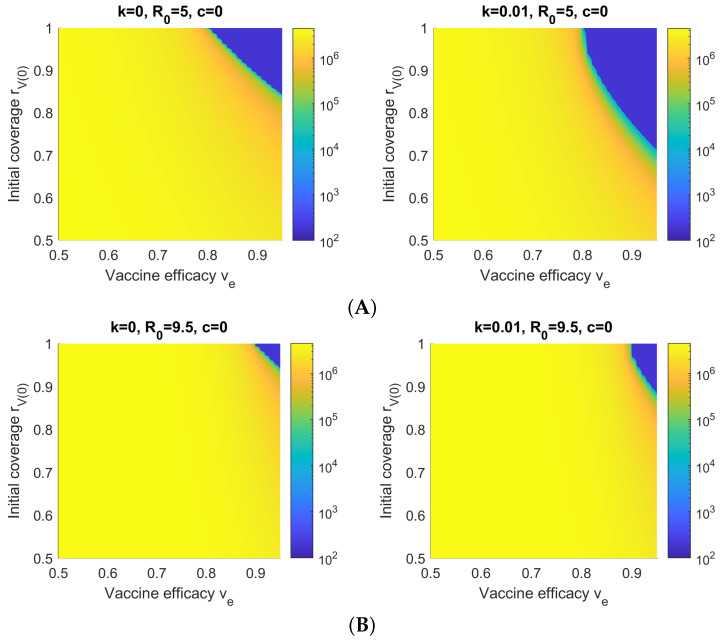
Cumulative hospitalizations/isolations $H_{c}\left(\infty \right)$ with respect to the initial vaccination coverage $r_{V}\left(0\right)$ and average vaccine efficacy $v_{e}$ for simulated (**A**) Delta and (**B**) Omicron outbreaks and no NPIs (c=0). The area of the blue region, representing a low number of total infections/hospitalizations, increases with *k*, the vaccination rate.

**Figure 7 viruses-14-01482-f007:**
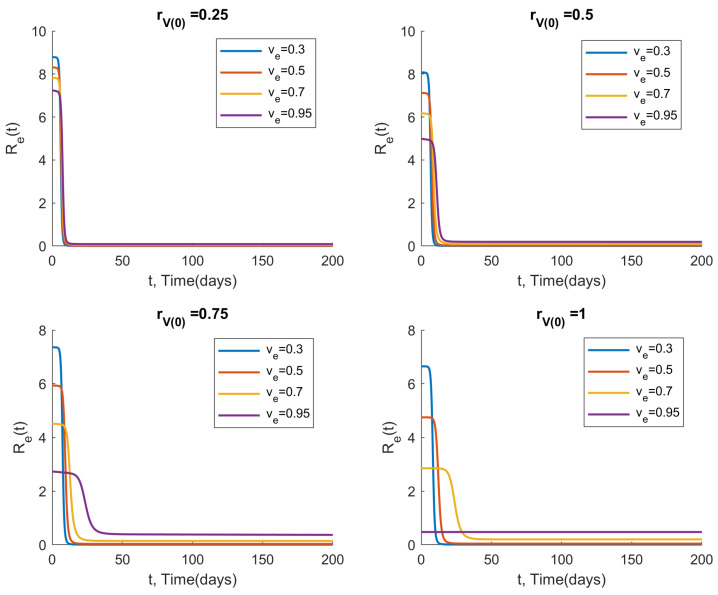
Effective reproduction number $R_{e}\left(t\right)$ with respect to time *t* for a simulated Omicron outbreak, with initial vaccination coverage ratio $r_{V(0)}$, vaccine efficacy $v_{e}$, a vaccination rate of k=0.002, and no NPIs (c=0). Note that the lowest vaccine efficiencies result in the lowest $R_{e}\left(t\right)$ for large *t*, as the supply of susceptible individuals becomes depleted.

**Figure 8 viruses-14-01482-f008:**
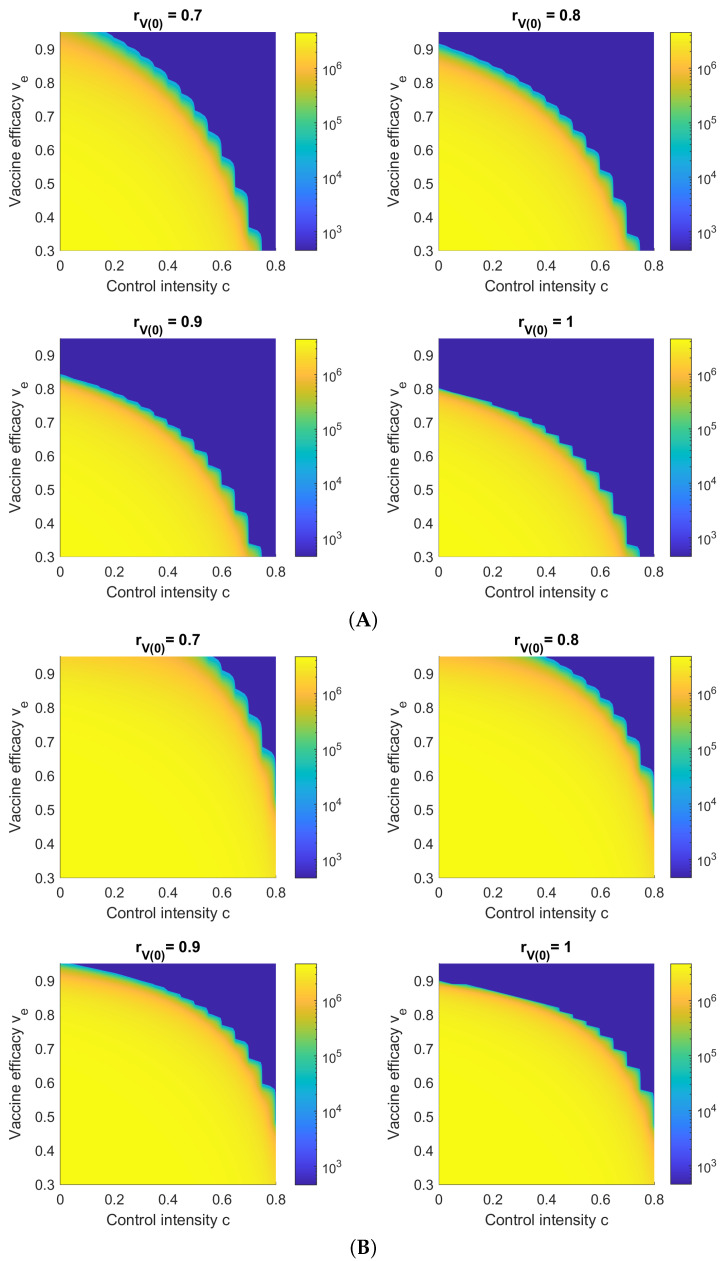
Cumulative hospitalizations/isolations $H_{c}\left(\infty \right)$ with respect to the vaccine efficacy $v_{e}$ and NPI control intensity *c* for simulated (**A**) Delta and (**B**) Omicron outbreaks and no NPIs (c=0). The area of the blue region, representing a low number of total infections/hospitalizations, increases with *k*, the vaccination rate.

**Figure 9 viruses-14-01482-f009:**
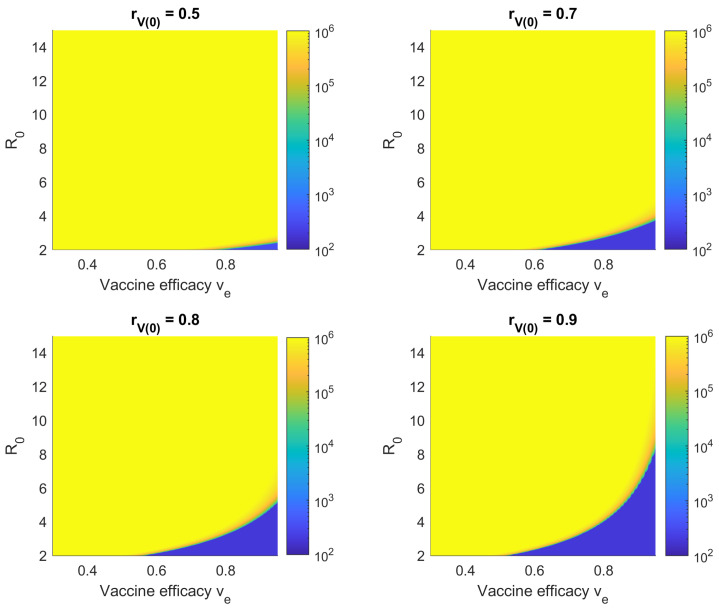
Cumulative hospitalizations and isolations $H_{c}\left(\infty \right)$ with respect to $R_{0}$ and the vaccine efficacy $v_{e}$ for a simulated epidemic outbreak and no NPIs (c=0). The area of the blue region, representing lower total infections/hospitalizations, increases with respect to $r_{V(0)}$, the initial vaccination coverage.

**Figure 10 viruses-14-01482-f010:**
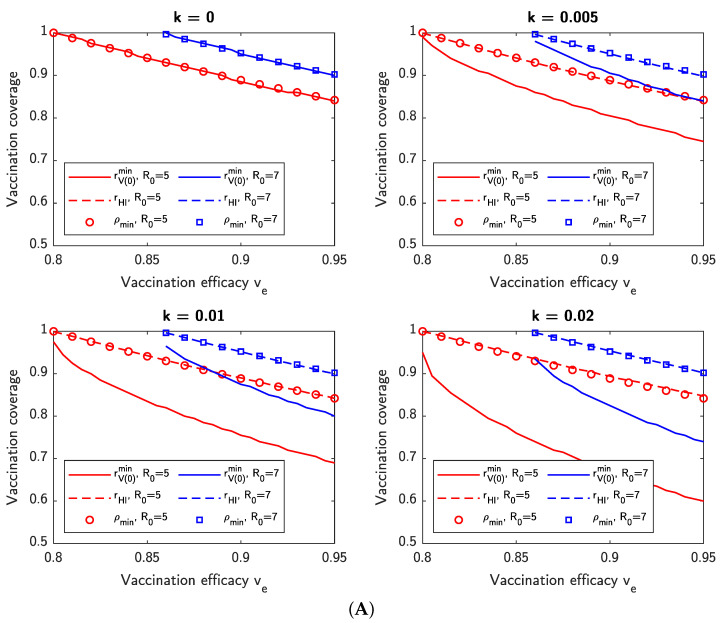
Minimum vaccination threshold required such that the cumulative number of hospitalizations and isolations is less than (**A**) 0.01N or (**B**) 0.1N, for different vaccine efficacies $v_{e}$ and vaccination rates *k*, and no NPIs (c=0). The theoretical herd immunity threshold $\rho_{\min}$ is also shown. It is demonstrated that for the 0.01N case, the proportion of vaccinated individuals when herd immunity is reached (the *dynamic vaccination threshold*) is approximately equal to the theoretical herd immunity threshold. On the other hand, for the 0.1N case, the dynamic vaccination threshold can be much less than the theoretical herd immunity threshold, with post-infection *natural immunity* becoming a major factor towards achieving herd immunity.

**Table 1 viruses-14-01482-t001:** List of symbols and parameters for the SVEIHR model depicted in Figure 1.

Symbol	Definition
*N*	Population size
ε	Transmission rate of exposed individuals, unvaccinated targets
α	Transmission rate of infected individuals, unvaccinated targets
$v_{e}$	(Mean) vaccine efficacy
$\varepsilon_{V}$	Transmission rate of exposed individuals, vaccinated targets (refer to Definition 1)
$\alpha_{V}$	Transmission rate of infected individuals, vaccinated targets (refer to Definition 1)
β	Rate of exposed individuals becoming symptomatic
γ	Hospitalization/isolation rate of infected individuals. Our model does not differentiate between hospitalization and non-clinical isolation of confirmed cases.
δ	Recovery rate of exposed, infected, or hospitalized/isolated individuals
*k*	Vaccination rate of susceptible individuals
$r_{V\left(0\right)}$	Ratio of vaccinated individuals at time 0 (i.e., V(0)/N, where *N* is the total population)
*c*	Control intensity of non-pharmaceutical interventions (NPIs), where c=0 denotes no control, and c=1 denotes complete cessation of disease transmission

**Table 2 viruses-14-01482-t002:** Initial values for the second and third waves of COVID-19 in Hong Kong with vaccination added.

Initial Value	Second Wave	Third Wave
V(0)	$r_{V(0)}N$	$r_{V(0)}N$
E(0)	9	9
I(0)	1	1
H(0)	0	0
R(0)	72	1193

*N* = 7,394,700, *S* (0) = *N* − *V* (0)− *E* (0) − *I* (0) − *H* (0) − *R* (0).

## Data Availability

All data included in this study are available upon request by contact with the corresponding author.

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
