# Peer review of "Modeling the Impact of Vaccination on COVID-19 and Its Delta and Omicron Variants"

_viruses, 2022, doi:10.3390/v14071482_

Round 1

Reviewer 1 Report

This manuscript proposed the SVEIHR model, and used this model to study the effects of vaccines, delta variant, Omicron variant and basic reproduction number on transmission dynamics. Taking Hong Kong as an example, this work points out whether herd immunity can be achieved and simulates the number of hospitalizations / infections under the influence of two variants. This work also found an interesting phenomenon, that is, the phenomenon of "drift" of the herd immunity threshold. In general, the workload of this work is relatively rich, and the research results are meaningful and generalizable.

I have the following comments:
1) Suppose there is a new type of virus with stronger infectivity, what is the prediction result of the model?

2) How to explain the physical meaning of the last two figures (figures 16 and 17)?

3) How can the SVEIHR model be used to help Hong Kong explain the epidemic situation in the first half of this year?

Reviewer 2 Report

In "Modeling the Impact of Vaccination on COVID-19 and its Delta and Omicron Variants" the authors present new research results dedicated to a vibrant and important subject, in particular related to halting epidemic spreading with vaccination in the face of different COVID-19 variants. 

I have very much enjoyed reading this paper. I find it comprehensive and clearly written, and introducing new, timely, and important results that will surely also inspire future research along these lines. The introduction is also quite comprehensive and informative. For these reasons, I am in favor of publication subject to the following revisions.

1) I would encourage the author to shorten the abstract and focus more on the key results. As it is, the abstract is quite verbose and the main messages dilute in such a style.

2) It would also improve the paper if the figure captions would be made more self-contained. In addition to what is shown for which parameter values, one could also consider a sentence or two saying what is the main message of each figure.

3) The presentation is not fully in keeping with an interdisciplinary readership. Too much technical details and mathematics is presented without much guidance of the reader through what is shown and why. This needs improvement for better clarify of the presentation for Viruses.

4) The introduction should be improved by referring also to closely related recent research regarding vaccination and similar models: Risk assessment of COVID-19 epidemic resurgence in relation to SARS-CoV-2 variants and vaccination passes, Tyll Krüger et al., Commun. Med. 2, 23 (2022) and Socio-demographic and health factors drive the epidemic progression and should guide vaccination strategies for best COVID-19 containment, Rene Markovič, et al., Results Phys. 26, 104433 (2021). These are closely related, and in general much more has been done in the past two years on this subject as the introduction currently gives credit to.

5) Also, it would be very useful if the authors would make their source code available as supplementary material. This would promote the usage of the proposed model and allow also others to take advantage of this research, and also to allow them to reproduce the results.

If a revision will be granted, I will be happy to review the manuscript again.
